# A Federated Transfer Learning Framework Based on Heterogeneous Domain Adaptation for Students' Grades Classification

**Bin Xu *** 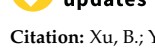**, Sheng Yan, Shuai Li and Yidi Du**

School of Computer Science and Engineering, Northeastern University, Shenyang 110169, China
*   Correspondence: xubin@mail.neu.edu.cn

**Abstract:** In the field of educational data mining, the classification of students' grades is a subject that receives widespread attention. However, solving this problem based on machine learning algorithms and deep learning algorithms is usually limited by large datasets. The privacy problem of educational data platforms also limits the possibility of building an extensive dataset of students' information and behavior by gathering small datasets and then carrying out the federated training model. Therefore, the balance of educational data and the inconsistency of feature distribution are the critical problems that need to be solved urgently in educational data mining. Federated learning technology enables multiple participants to continue machine learning and deep learning in protecting data privacy and meeting legal compliance requirements to solve the data island problem. However, these methods are only applicable to the data environment with common characteristics or common samples under the alliance. This results in domain transfer between nodes. Therefore, in this paper, we propose a framework based on federated transfer learning for student classification with privacy protection. This framework introduces the domain adaptation method and extends the domain adaptation to the constraint of federated learning. Through the feature extractor, this method matches the feature distribution of each party in the feature space. Then, labels and domains are classified on each side, the model is trained, and the target model is updated by gradient aggregation. The federated learning framework based on this method can effectively solve the federated transfer learning on heterogeneous datasets. We evaluated the performance of the proposed framework for student classification on the datasets of two courses. We simulated four scenarios according to different situations in reality. Then, the results of only source domain training, only target domain training, and federated migration training are compared. The experimental results show that the heterogeneous federated transfer framework based on domain adaptation can solve federated learning and knowledge transfer problems when there are little data at the data source and can be used for students' grades classification in small datasets.

**Keywords:** federated learning; transfer learning; students' grades classification; domain adaptation; educational data mining

## 1. Introduction

With the development of artificial intelligence, especially the emergence of neural networks, artificial intelligence technology driven by big data has played its role in many complex fields, such as automatic driving, medical care, and finance. People want this technology to be realized in all walks of life. However, the actual situation is very disappointing. Except for a few limited fields, many industries have limited data and poor data quality, which is insufficient to support the implementation of artificial intelligence technology [1]. Even if there are many data in more application fields, these data are distributed among different data sources and marked unevenly. For example, there are 537,100 schools of all levels and types in China, with 289 million students. These students' data are stored

in different schools and on various online education platforms. These data sources are not connected to each other, and the data exist in the form of isolated islands. It is almost impossible to integrate these data, or the cost required is enormous.

On the other hand, with the further development of big data, data privacy and security have become a worldwide trend. Because of IoT devices' increasing storage and computing capabilities, it is common to save data and models on local devices [2]. Federated learning is a new frontier field [1,3,4]. Google has launched a federated learning system [3–5], in which a federated of distributed participants updates the global machine learning model and their data are stored locally. Their framework requires all contributors to share the same feature space. On the other hand, secure machine learning for partitioning data in feature space is also studied [6–8], which finds that, in machine learning, each participant can make joint modeling with the help of other parties' data. However, all parties do not need to share data resources. They can conduct federated data training and establish a shared machine learning model when the data do not come out locally. Federated learning enables multiple participants to continue machine learning and deep learning in protecting data privacy and meeting legal compliance requirements to solve the data island problem. Federated learning technology can be used among mobile terminals, wearable devices, and different platforms to realize a training machine learning model that pays more attention to privacy and efficiency. However, when one party collects a lot of unlabeled data, even when the features are statistically different from those collected by the other party, domain shift will arise [9]. This makes the model obtained by federated learning and training unable to be extended to new equipment. For example, in the field of education, two different online education platforms save local student data. One platform may collect more student behavior logs, while the other platform saves students' personal information.

In the field of educational data mining, the classification of students' grades is usually limited by the size of datasets and the number of data features. Moreover, because of data privacy protection, each data platform cannot carry out practical federated training. Therefore, it is an urgent problem to solve the imbalance of data and the inconsistency of characteristics among data owners to train student models.

This paper proposes a framework based on federated transfer learning for students' grades classification with privacy protection. In this paper, heterogeneous domain adaptive technology is introduced into federated transfer learning, and knowledge transfer between different nodes is solved by exchanging encryption parameters between data. In this framework, each data owner uses the network of label classifier and domain classifier for training and aligns the features of all parties in the new feature space for transfer learning. Furthermore, according to the actual situation, we also simulate the following four scenarios by constraining different characteristics and data scales:

- Scenario 1: Both Party A and Party B have the same feature distribution and data scales;
- Scenario 2: Party A and Party B have the same characteristics but different data scales;
- Scenario 3: Party A and Party B have different characteristics but the same data scales;
- Scenario 4: Both Party A and Party B have different feature distribution and data scales.

We conducted experiments on datasets from two middle schools in Portugal and obtained the training results of student models in these four situations. Experimental results show that both parties with different data feature spaces and inconsistent data scales can effectively transfer knowledge under this framework. It can help the target data party with missing data labels or fewer feature data to train and improve the generalization ability of the model of main data participants.

The rest of this paper is organized as follows: In Section 2, we outline related work. In Section 3, we present the details of the model proposed in this paper. Then, in Section 4, we describe the experimental details. The analysis of the experimental results is in Section 5. Finally, in Section 6, we summarize our work and present some ideas for future work.

## 2. Related Work

### 2.1. Domain Adaptation

Domain adaptation is an active research topic in many fields of artificial intelligence, including machine learning, natural language processing, and computer vision. The methods of early domain adaptation mainly focus on constructing domain-invariant features. This can be achieved by the weight and selection mechanism of features [10,11] or by learning the explicit feature transformation that aligns the distribution of source domain and target domain [12–14]. In recent work, deep neural networks have been used to learn powerful representation and perform unsupervised domain adaptation [15]. Revgrad [16] uses a domain classification network, which aims to distinguish between source and target embedding. The goal of the feature extraction network is to produce embedding that maximizes the loss of domain classifiers and minimizes the loss of tag prediction. This is achieved by negating the gradient from the domain classification network.

Driven by GAN [17], many transfer learning methods are based on the assumption that a good feature representation contains almost no specific information about the original domain of an instance. For example, Ganin et al. proposed a depth structure called Domain-Adversarial Neural Network (DANN) for domain adaptation [16,18]. Its architecture consists of a feature extractor, tag predictor, and domain classifier. DANN can be trained by inserting a particular gradient reversal layer (GRL). After the training of the whole system, the feature extractor learns the deep features of instances, and the output classifies the prediction tags of unmarked target domain instances.

In the field of domain adaptation, some researchers have improved and applied domain adaptation technology. Cao et al. put forward an adaptive model with semantic consistency (ADASC), which gradually and effectively aligns the distinguishing features across domains using the class hierarchy relationship between domains [19]. Hu et al. developed a framework DATSING based on transfer learning, which effectively utilized the potential representation of cross-domain time series to enhance target domain prediction [20]. Su et al. introduced the task of continuous domain adaptation in machine reading comprehension [21].

There are some other related impressive works. Zeng et al.' s work puts forward a unified framework for adaptation in adversarial fields [15]. Shen et al. used Wasserstein distance for domain adaptation [22]. Hoffman et al. adopted cycle-consistency loss to ensure the consistency of structure and semantics [23]. Long et al. put forward the condition domain adversarial network, which uses the condition domain discriminator to assist the adversarial adaptation [24]. Zhang et al. adopted the symmetrical design of source classifier and target classifier [25]. Zhao et al. used a domain confrontation network to solve the problem of multi-source transfer learning [26]. Yu et al. proposed a dynamic adversarial adaptation network [27]. Chen et al. proposed a reinforcement learning (DARL) framework combined with adversarial learning for partial domain adaptation [28]. The DARL framework adopts the deep Q-learning method to learn the strategy by approximating the action value function, so that the agent can make a choice decision. In order to minimize the domain differences in cross domain knowledge transfer, Chen et al. used self similarity consistency (SSC) constraints to match the deep feature structures in the source and target domains, rather than aligning the global distribution statistics across domains [29]. Tran et al. proposed a generative adversary domain adaptive (GADA) algorithm, which significantly improved the process of distinguishing feature extraction by injecting additional classes and using the generated samples for training [30]. Guo et al. proposed a new distribution calibration method by learning an adaptive weight matrix between new samples and base classes, which built on the Hierarchical Optimal Transport (H-OT) framework by minimizing the high-level OT distance between the new sample and the base class to convey the adaptive weight information of the base class [31]. In the multi-source domain adaptation task, Deng et al. proposed a two-layer optimization-based robust object training (BORT2) [32]. BORT2 first learns a labeling function using source and target data, and then trains a noise-robust model only on the pseudo-labeled target

domain. Noise-robust models exploit feature uncertainty to detect label noise and mitigate its negative effects.

To sum up, although there are many kinds of research in the field of domain adaptation in recent years, this research is all based on typical scene training. Their data are stored on the same server, which leads to incompatibility under the condition of federated learning. Therefore, our research hopes to achieve domain adaptation under the framework of federated learning.

### 2.2. Federated Learning

Google recently proposed the concept of federated learning [3–5]. Their main idea is to build a machine learning model based on datasets distributed on multiple devices while preventing data leakage. Recent improvements focus on overcoming statistical challenges [2,33] and improving the security of federated learning [34,35], and there are some research efforts to make federated learning more personalized [36]. All of the above work focuses on device federated learning, involving distributed mobile user interaction and communication cost, unbalanced data distribution, and device reliability in large-scale distribution are some of the main factors of optimization. To extend the concept of federated learning to cover collaborative learning scenarios among organizations, Yang et al. expanded the original federated learning into the general idea of all decentralized collaborative machine learning technologies for privacy protection [1]. They made a comprehensive overview of federated learning and federated transfer learning technologies. At the same time, they further investigated the related security foundation and discussed the relationship with other associated fields, such as multi-agent theory and privacy protection data mining. Cheng et al. proposed a new lossless privacy protection algorithm secureboost, which is used to train high-quality tree enhancement models when the training data are kept secret on multiple parties [7]. Chandra et al. proposed a new distributed machine learning method, called splitfed. Due to network fragmentation, it provides a higher level of privacy than federated learning. It is suitable for machine learning (ML) with low computational resources, fast training, and analysis of private and sensitive data [37]. Mahmood et al. studied the feasibility of data and model poisoning attacks under the blockchain based FL system built next to the Ethereum network and the traditional FL system (no blockchain) [38]. They proposed a transparent incentive mechanism to fill the knowledge gap, which can encourage good behavior among participating decentralized nodes, avoid common problems, and provide knowledge for FL security literature by studying the current FL system.

In addition, the time required for computing is also an important factor. Federated learning (FL) is largely affected by the heterogeneity of clients. The training speed of clients is inconsistent. In order to reduce the impact of slow clients, Cox et al. proposed a new FL algorithm using model freezing and unloading, which uses a simple, fast, and scalable scheduling algorithm to periodically organize model unloading between clients [39]. To reduce the communication and computation delays of model parameters, Liu et al. proposed a new iterative algorithm and minimized the total model parameter communication and computation delays by optimizing the number of local iterations and edge iterations [40].

Recent research has focused on transferring knowledge from decentralized nodes to new nodes with different data domains. Liu et al. introduced new technology and framework called federated transfer learning (FTL) to improve the statistical model under the data alliance. The alliance allows knowledge sharing without compromising users' privacy and allows supplementary knowledge to be transmitted in the network [8]. Therefore, the target domain can build a more flexible and robust model using rich tags from the source domain. Peng et al. put forward a principled method to solve the federated domain adaptation problem, which aims to make the representations learned between different nodes consistent with the data distribution of target nodes [41].

However, we do not think Peng's method extends to the constraint of federated learning. It cannot be applied to the general environment of federated learning. Therefore, in this

paper, we combine domain adaptation with secure federated learning and propose a new federated transfer learning framework based on domain adaptation to solve knowledge transfer in federated transfer learning.

## 3. Method

In this section, we introduce the details of our proposed method. The network we use is a deep neural network. The network structure we propose is a deep neural network with three layers whose structure is shown in Figure 1. The input data are passed through a feature extractor to match the feature distribution of each party in the feature space. Then, the labels and domains are classified on each side, the model is trained, and the target model is updated by gradient aggregation.

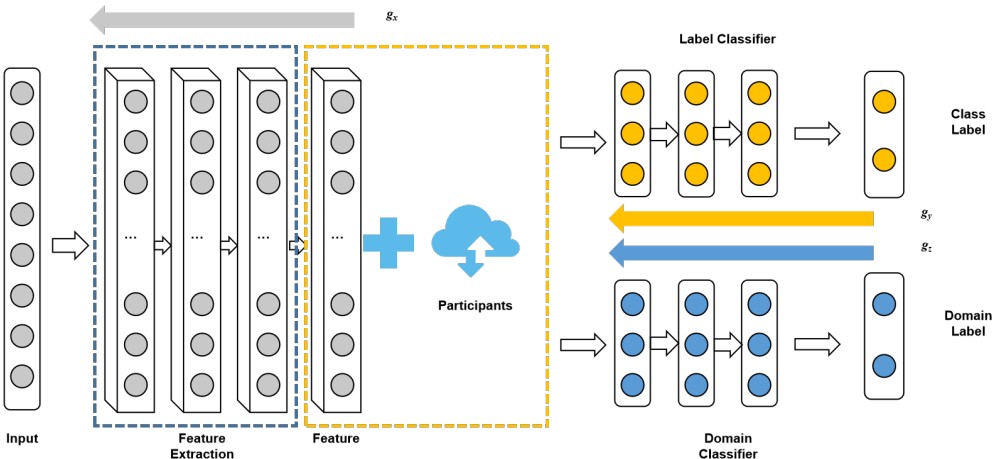

**Figure 1.** Structure diagram of a neural network in the experiment.

### 3.1. Problem Definition

In the actual educational data mining, we often encounter the situation of small data scale and lack of data features and need to manually mark them. However, due to the data privacy protection, in many cases, the data owners can not carry out collaborative training. To solve this problem, we simulated a federated training scenario between Party A and Party B.

Consider two data owners, a source domain $D_A = \{(x_i^A, y_i^A, z^A)_{i=1}^n\}$ where $x_i^A \in \mathbb{R}^n$, $y_i^A \in \{0,1\}$ is the $i$th label and $z^A = 1$ is the domain label, and a target domain $D_B = \{(x_i^B, z^B)_{i=1}^{n'}\}$ where $x_i^B \in \mathbb{R}^{n'}$, $z^B = 0$ is the domain label. $n, n'$ represents the number of samples of the source domain and target domain, respectively. The total number of samples is $N = n + n'$.

In our definition, Party A and Party B represent two platforms with educational data. Among them, Party A has complete data, label information, and class labels. Party A is the leading party in federated learning and the owner of the source domain data in federated transfer learning. Party B only has some data and class labels and has no data labels. Play the role of the customer in federal migration learning. Party A and Party B use the knowledge gained from federal migration learning to build a model and predict their users.

Our goal is to establish a transfer learning model, accurately predict the category of students of Party B, and improve the classification effect of Party A. At the same time, their data will not be exposed. In this paper, domain adaptation is introduced into federated migration learning, which minimizes and maximizes the classification errors of the source domain, and then exchanges knowledge between the source domain and the target domain. Then, this federated learning problem can be formulated as follows:

$$\underset{w \in \mathbb{R}^N}{\text{minimize}} \mathcal{L}_1(w, D) = \frac{1}{N} \sum_{i=1}^{N} f(w; x_i, y_i) \tag{1}$$

$$\underset{\boldsymbol{w} \in \mathbb{R}^N}{\text{maximize}} \mathcal{L}_2(\boldsymbol{w}, D) = \sum_{i=1}^{N} \gamma(\boldsymbol{w}; x_i, z_i) \tag{2}$$

where $\boldsymbol{w}$ is the parameter of the model, $f$ represents the loss function of the label classifier, and $\gamma$ represents the loss function of the domain classifier.

Our goal is to exchange only intermediate parameters to achieve the following goals on privacy protection:

$$\underset{\boldsymbol{w} \in \mathbb{R}^N}{\text{minimize}} \mathcal{L} = \mathcal{L}_1(\boldsymbol{w}, D) - \mathcal{L}_2(\boldsymbol{w}, D) \tag{3}$$

*3.2. Approach Statement*

We assume that there are two data owners *Party A & B* training model $A$ and model $B$. Firstly, the model needs feature extraction to generate two implicit representations $u_i^A = '_A(x_i^A)$ and $u_i^B = '_B(x_i^B)$, where $u^A \in \mathbb{R}^{N_A \times d}$ and $u^B \in \mathbb{R}^{N_B \times d}$, and $d$ represent the dimensions of the hidden layer. $\varphi(x; w, b) = \frac{1}{1 + \exp(-w^\mathsf{T} x)}$ is the feature transformation function, where matrix-vector pair $(w, b) \in \mathbb{R}^N \times \mathbb{R}$. The neural network $\varphi_A$ and $\varphi_B$ transform their features, and project the features of *Party A* and *Party B* into a common feature subspace, where they can exchange knowledge.

Then, the source domain data of feature space is classified by the prediction layer, and the correct labels are separated as far as possible. The prediction function $\psi(u_i^B) = \psi(u_1^A, y_1^A ... u_{N_A}^A, y_{N_A}^A, u_j^B)$,

where $\psi(\mathbf{a}) = \left[ \frac{\exp(a_i)}{\sum_{j=1}^{|\mathbf{a}|} \exp(a_j)} \right]_{i=1}^{|\mathbf{a}|}$. Thus, we obtain the first optimization goal:

$$\underset{\Theta^x, \Theta^y}{\text{argmin}} \mathcal{L}_1 = \frac{1}{n} \sum_{i=1}^{T} \ell_1 \left( y_i^A, \psi\left( u_i^B \right) \right) \tag{4}$$

where $\Theta^x, \Theta^y$ are training parameters of layers $\varphi_A, \psi$. $\ell_1$ is classification loss, using the negative log-probability of the correct labels.

The network also needs a regularization term, that is, a metric criterion so that the source domain and the target domain in the feature space are as close as possible. We use H-divergence to measure [42–44]:

$$\hat{d}_{\mathcal{H}}(S, T) = 2 \left( 1 - \min_{\eta \in \mathcal{H}} \left[ \frac{1}{n} \sum_{i=1}^{n} I[\eta(\mathbf{x}_i) = 0] \right. \right. \\ \left. \left. + \frac{1}{n'} \sum_{i=n+1}^{N} I[\eta(\mathbf{x}_i) = 1] \right] \right) \tag{5}$$

where $\mathcal{H}$ is a hypothetical class, binary classifiers $\eta : X \to \{0, 1\}$.

According to formula (5), the empirical H-divergence between the source domain and the target domain can be calculated by the following formula:

$$\hat{d}_{\mathcal{H}} \left( D^A, D^B \right) = 2 \left( 1 - \min_{\eta \in \mathcal{H}} \left[ \frac{1}{n} \sum_{i=1}^{n} I\left[ \eta\left( \omega\left( u_i^B \right) \right) = 0 \right] \right. \right. \\ \left. \left. + \frac{1}{n'} \sum_{i=n+1}^{N} I\left[ \eta\left( \omega\left( u_i^B \right) \right) = 1 \right] \right] \right) \tag{6}$$

It is usually difficult to calculate $\hat{d}_{\mathcal{H}}(D^A, D^B)$. According to the inspiration of Ben-David's research [42,43], we can define the approximate solution of the classification problem of the source domain and target domain, that is, domain classifier. It classifies data in feature space and tries to distinguish which domain the data comes from. Domain

classifier $\omega(u_j^B) = \frac{1}{1+\exp(-u_j^B)}$, and we can make the feature distribution of the data in the source domain and the target domain as close as possible by approximately calculating the empirical H-divergence between the two samples. We can obtain the second optimization target:

$$\underset{\Theta^y, \Theta^z}{\mathrm{argmax}} \mathcal{L}_2 = -\frac{1}{n} \sum_{i=1}^{n} \ell_2\left(z_i^A, \omega\left(u_i^B\right) = 0\right)$$
$$-\frac{1}{n'} \sum_{i=n+1}^{N} \ell_2\left(z_i^A, \omega\left(u_i^B\right) = 1\right) \tag{7}$$

where $\Theta^y, \Theta^z$ are training parameters of layers $\psi, \omega$. $z_i$ is the domain classification label. $\ell_2$ is the loss of domain classification layer, and the loss of feature alignment is calculated by cross entropy.

The final objective function is:

$$\underset{\Theta^x, \Theta^y, \Theta^z}{\mathrm{argmin}} \mathcal{L} = \frac{1}{n} \sum_{i=1}^{n} \ell_1\left(y_i^A, \psi\left(u_i^B\right)\right)$$
$$+ \lambda \left(\frac{1}{n} \sum_{i=1}^{n} \ell_2\left(z_i^A, \omega\left(u_i^B\right) = 0\right)\right.$$
$$+ \frac{1}{n'} \sum_{i=n+1}^{N} \ell_2\left(z_i^A, \omega\left(u_i^B\right) = 1\right)\Big) \tag{8}$$

where $\lambda$ is the weight parameters. For $i \in \{A, B\}$, the stochastic gradients can be computed as the following:

$$g_x^i = \frac{\partial \mathcal{L}_1^i}{\partial \Theta_i^x} - \lambda \frac{\partial \mathcal{L}_2^i}{\partial \Theta_i^x} \tag{9}$$

$$g_y^i = \frac{\partial \mathcal{L}_1^i}{\partial \Theta_i^y} \tag{10}$$

$$g_z^i = \frac{\partial \mathcal{L}_2^i}{\partial \Theta_i^z} \tag{11}$$

### 3.3. Federated Domain Adaptation

In order to realize secure computation without exchanging data, additively homomorphic encryption is adopted. We define a number $x$, which becomes $[\![x]\!]$ after additively homomorphic encryption. According to the properties of homomorphic encryption, for any two numbers $x$ and $y$, $[\![x]\!] + [\![y]\!] = x + y$ and $x \times [\![y]\!] = [\![x]\!] \times [\![y]\!]$ is valid. However, the loss function and gradient can not be directly calculated by additively homomorphic encryption. In order to solve this problem, we use the second-order Taylor approximation for the logistic loss

$$\ell(y, \varphi) \approx \ell(y, 0) - \frac{1}{2} y\varphi + \frac{1}{8} y^2 \varphi^2 \tag{12}$$

and the gradient is

$$\frac{\partial \ell}{\partial \varphi} = -\frac{1}{2} y + \frac{1}{4} y^2 \varphi \tag{13}$$

In this way, we obtain the encrypted loss and gradient.

$$[\![\mathcal{L}_1]\!] = \frac{1}{n} \sum_{i=1}^{n} \left([\![\ell_1(y_i^A, 0)]\!] - \frac{1}{2} y_i^A [\![\psi\left(u_i^B\right)]\!]\right.$$
$$+ \frac{1}{8} y_i^{A^2} [\![\psi\left(u_i^B\right)' \psi\left(u_i^B\right)]\!]\Big) \tag{14}$$

$$
\begin{aligned}
[\![\mathcal{L}_2]\!] = {} & -\frac{1}{n}\sum_{i=1}^{n}\left([\![\ell_2(z_i^A,0)]\!] - \frac{1}{2}z_i^A[\![\omega\left(u_i^B\right)]\!]\right.\\
& \left. +\frac{1}{8}z_i^{A^2}[\![\omega\left(u_i^B\right)'\omega\left(u_i^B\right)]\!]\right)\\
& -\frac{1}{n'}\sum_{i=n+1}^{n'}\left([\![\ell_2(z_i^A,0)]\!] - \frac{1}{2}z_i^A[\![\omega\left(u_i^B\right)]\!]\right.\\
& \left. +\frac{1}{8}z_i^{A^2}[\![\omega\left(u_i^B\right)'\omega\left(u_i^B\right)]\!]\right)
\end{aligned}
\tag{15}
$$

$$
\begin{aligned}
\left[\!\!\left[\frac{\partial \mathcal{L}_1^i}{\partial \Theta_i}\right]\!\!\right] = {} & \frac{1}{n}\sum_{i=1}^{n}\frac{\psi\left(u_i^B\right)'\psi\left(u_i^B\right)}{\partial u_i^B}\left[\!\!\left[\left(\frac{1}{8}y_i^{A^2}\psi\left(u_i^B\right)'\psi\left(u_i^B\right)\right)\right]\!\!\right]\frac{\partial u_i^B}{\partial \Theta_i}\\
& +\frac{1}{n}\sum_{i=1}^{n}[\![\frac{1}{2}y_i^A\psi\left(u_i^B\right)]\!]\frac{\partial \psi\left(u_i^B\right)}{\partial u_i^B}\frac{\partial u_i^B}{\partial \Theta_i}
\end{aligned}
\tag{16}
$$

$$
\begin{aligned}
\left[\!\!\left[\frac{\partial \mathcal{L}_2^i}{\partial \Theta_i}\right]\!\!\right] = {} & -\frac{1}{n}\sum_{i=1}^{n}\frac{\omega\left(u_i^B\right)'\omega\left(u_i^B\right)}{\partial u_i^B}\left[\!\!\left[\left(\frac{1}{8}z_i^{A^2}\omega\left(u_i^B\right)'\omega\left(u_i^B\right)\right)\right]\!\!\right]\frac{\partial u_i^B}{\partial \Theta_i}\\
& -\frac{1}{n}\sum_{i=1}^{n}[\![\frac{1}{2}z_i^A\omega\left(u_i^B\right)]\!]\frac{\partial \omega\left(u_i^B\right)}{\partial u_i^B}\frac{\partial u_i^B}{\partial \Theta_i}\\
& -\frac{1}{n'}\sum_{i=1}^{n}\frac{\omega\left(u_i^B\right)'\omega\left(u_i^B\right)}{\partial u_i^B}\left[\!\!\left[\left(\frac{1}{8}z_i^{A^2}\omega\left(u_i^B\right)'\omega\left(u_i^B\right)\right)\right]\!\!\right]\frac{\partial u_i^B}{\partial \Theta_i}\\
& -\frac{1}{n'}\sum_{i=1}^{n}[\![\frac{1}{2}z_i^A\omega\left(u_i^B\right)]\!]\frac{\partial \omega\left(u_i^B\right)}{\partial u_i^B}\frac{\partial u_i^B}{\partial \Theta_i}
\end{aligned}
\tag{17}
$$

Then, update the parameters according to the gradient direction

$$
\Theta^x \leftarrow \Theta^x - \eta g_x
\tag{18}
$$

$$
\Theta^y \leftarrow \Theta^y - \eta g_y
\tag{19}
$$

$$
\Theta^z \leftarrow \Theta^z - \eta\lambda g_z
\tag{20}
$$

where $\eta$ is the learning rate. With these parameters, we now design a federated Domain Adaptation network for solving the Domain Adaptation problem. As shown in Figure 2. First, Party A and Party B send public keys to each other, then initialize parameters $\Theta^x, \Theta^y, \Theta^z$, calculate implicit representations $u^A$ and $u^B$, and then calculate, encrypt, and exchange intermediate results $g$ and $\mathcal{L}$.

To prevent exposure of the gradients of A and B, A and B further mask each gradient with encrypted random values. Then, they send the gradient and loss of the mask to each other and locally decrypt these values. Once the loss convergence condition is met, Party A can send a termination signal to Party B. Otherwise, they will unmask the gradients, update the model parameters with their gradients, and then continue the next iteration. Algorithm 1 summarizes the process.

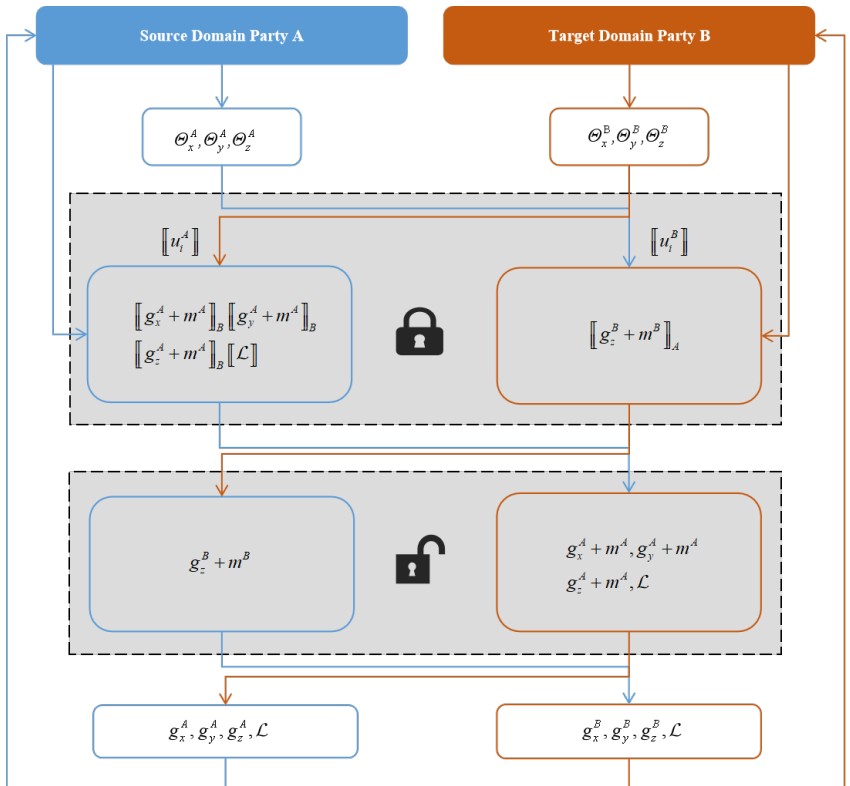

**Figure 2.** System Architecture of Federated Domain Adaptation.

---

**Algorithm 1** Federated domain adaptation

---

**Input:** learning rate: $\eta$, adaptation parameter: $\lambda$
**Output:** Model parameters $\Theta^x, \Theta^y, \Theta^z$ *Initialization:Patty A&B initialize* $\Theta^x, \Theta^y, \Theta^z$
    **while** Convergence conditions is not met **do**
        **for all** each iteration $k = 1, 2, 3...$ **do**
          Part A&B compute: $u_i^A, u_i^B$
          A&B create a random mask: $m^A, m^B$
          A compute and encrypt: $\mathcal{L}^A, g_x^A + m^A, g_y^A + m^A, g_z^A + m^A$ and send to B
          B compute and encrypt: $g_x^B + m^B, g_y^B + m^B, g_z^B + m^B$ and send to A
          A&B decrypt gradients and exchange, unmask and update model locally
              $\Theta^x \leftarrow \Theta^x - \eta g_x$
              $\Theta^y \leftarrow \Theta^y - \eta g_y$
              $\Theta^z \leftarrow \Theta^z - \eta \lambda g_z$
        **end for**
    **end while**
    **return** $\Theta^x, \Theta^y, \Theta^z$

---

*3.4. Security Analysis*

In this section, we aim to determine whether one party participating in federal training can learn the other party's data from the parameter set $H^k$ exchanged during training. Previous studies mainly introduced the leakage of the whole set of model parameters or gradient in the training process [45–47]. However, in our framework, the model's parameters are confidential, and only the intermediate results in the tag prediction layer may reveal information.

**Theorem 1.** *According to our definition, the federated transfer learning framework based on domain heterogeneous adaptation is secure, provided that the underlying encryption parameter sharing scheme is secure.*

**Proof.** In our framework, the model's parameters are confidential, and only the intermediate results in the label classifier layer may reveal information. In the domain classifier layer, each party can only receive the information of the domain but not the information of the data. Let $S_j$ be the set of data points sampled in the $t$-th iteration and $i_j$ be the $i$-th sample in the $j$-th iteration. $H_{i_j}$ refers to the contribution of the $i$-th sample to other parties. In the $j+1$-th iteration, we can obtain the weight update of the label classifier according to Equation (19):

$$\Theta_{j+1}^y = \Theta_j^y - \eta_j \left( \frac{1}{N_{S_j}} \sum_{i_j \in S_j} \left( -\frac{1}{2} y_{i_j} + \frac{1}{4} y_{i_j}^2 H_{i_j} \right) \right) \tag{21}$$

where $H_{i_j} = \varphi(x_{i_j})\Theta_j^y$. For any participant $k$ with unpublished datasets $D_k$ and training parameters $\Theta_k^y$ following our defined framework, there is an infinite solution of $\{x_{i_j}^k\}_{i \in S_{j}, j=0,1\cdots}$ that produces the same contribution set $\{H_{i_j}^k\}_{j=0,1\cdots}$. That is to say, no matter the number of iterations, the data of participant $k$ cannot be uniquely determined from its exchanged information $\{H_{i_j}^k\}_{j=0,1\cdots}$. This definition of security is consistent with the previous definition of security proposed in privacy protection and secure multiple computation of machine learning [48]. Under this definition, when some prior knowledge about data are known, the other party may exclude some alternative solutions but cannot infer accurate original data. This model provides a flexible trade-off between privacy and efficiency. □

**Theorem 2.** *According to our definition, the federated transfer learning framework based on domain heterogeneous adaptation is secure, provided that the homomorphic encryption scheme used is secure.*

**Proof.** In our scheme, all the learning and training data are encrypted by a homomorphic encryption mechanism. The parameters and intermediate data of the model are encrypted in the whole federated learning process. In each iteration, both sides will create a new random encryption mask $m$. This randomness and confidentiality also ensure the security of the whole training and learning process. Therefore, as long as the encryption scheme is secure, the federated learning framework we defined is also secure. □

## 4. Experiments

In this section, we mainly introduce the usage of our proposed framework and verify it by constraining different features, quantity, and the correlation between features on the datasets of two courses.

### 4.1. Datasets and Experimental Settings

The data used in the experiment come from the students' achievements in secondary education in two Portuguese schools. Data attributes include student achievement, demographic, social, and school-related characteristics collected using school reports and questionnaires. The two datasets provide the performance of two different subjects: mathematics and Portuguese. There are 396 students in the mathematics dataset and 650 students in the Portuguese dataset. We selected 16 attributes, which are described in Table 1. Students are classified in a binary system, and the G3 grades provide the classification labels of students. Students with scores greater than ten are classified into one category, while others are classified.

In our experiments, we set the depth of DNN layers as three in the model architecture. The learning rates are set to 0.0001, and batch size is set to 128. To avoid overfitting, we apply dropout on the DNN layers with a drop rate of 0.3.

**Table 1.** Preprocessed student-related variables and their descriptions

| No. | Attribute | Description (Domain) |
| --- | --- | --- |
| 1 | address | student's home address type (binary: "U"—urban or "R"—rural) |
| 2 | famsize | family size (binary: "LE3"—less or equal to 3 or "GT3"—greater than 3) |
| 3 | Medu | mother's education (numeric: 0—none, 1—primary education (4th grade), 2—5th to 9th grade, 3—secondary education or 4—higher education) |
| 4 | Fedu | father's education (numeric: 0—none, 1—primary education (4th grade), 2—5th to 9th grade, 3—secondary education or 4—higher education) |
| 5 | studytime | weekly study time (numeric: 1—<2 h, 2—2 to 5 h, 3—5 to 10 h, or 4—>10 h) |
| 6 | paid | extra paid classes within the course subject (Math or Portuguese) (binary: yes or no) |
| 7 | activities | extra-curricular activities (binary: yes or no) |
| 8 | higher | wants to take higher education (binary: yes or no) |
| 9 | internet | Internet access at home (binary: yes or no) |
| 10 | romantic | with a romantic relationship (binary: yes or no) |
| 11 | famrel | quality of family relationships (numeric: from 1—very bad to 5—excellent) |
| 12 | freetime | free time after school (numeric: from 1—very low to 5—very high) |
| 13 | goout | going out with friends (numeric: from 1—very low to 5—very high) |
| 14 | Dalc | workday alcohol consumption (numeric: from 1—very low to 5—very high) |
| 15 | Walc | weekend alcohol consumption (numeric: from 1—very low to 5—very high) |
| 16 | absences | number of school absences (numeric: from 0 to 93) |
| 17 | G1 | first period grade (numeric: from 0 to 20) |
| 18 | G2 | second period grade (numeric: from 0 to 20) |
| 19 | G3 | final grade (numeric: from 0 to 20, output target) |

### *4.2. Privacy Protection*

In order to ensure the privacy of the framework, the experiment simulates the distributed training scenario of multiple nodes, and the data of Party A and Party B are stored separately in each node to participate in the federated learning. In order to promote secure computing between the two parties, a semi-honest third-party server is introduced, and it is assumed that it does not collude with other participants. After learning and training, each participant can only hold model parameters associated with their characteristics. Therefore, when reasoning, both parties need to work together to generate the output. The third-party server does not obtain the data of all parties but only distributes the secret key and exchanges the encryption parameters.

### *4.3. Design*

In the experiment, according to the datasets size and data feature distribution of Party A and Party B, four scenarios were established to simulate the actual situation. They are as follows:

- Scenario 1: Both Party A and Party B have the same feature distribution and data scales;
- Scenario 2: Party A and Party B have the same characteristics but different data scales;
- Scenario 3: Party A and Party B have different characteristics but the same data scales;
- Scenario 4: Both Party A and Party B have different feature distribution and data scales.

To correspond to these four scenarios, we processed the data and selected some data and attributes. In addition, in the scene with different feature distribution, the correlation analysis of attributes is carried out to ensure that the correlation with domain invariant features does not exceed the threshold. We carry out a factor analysis on the features contained in the datasets. The calculated factor scores are shown in Table 2, indicating eight principal components in total.

**Table 2.** Results of factor analysis

|  | Factor1 | Factor2 | Factor3 | Factor4 | Factor5 | Factor6 | Factor7 | Factor8 |
|---|---|---|---|---|---|---|---|---|
| address | −0.0174 | −0.04261 | −0.02972 | 0.55018 | −0.03883 | 0.02952 | 0.06146 | −0.11739 |
| famsize | 0.05361 | −0.12259 | 0.00306 | 0.04264 | −0.02741 | −0.00522 | 0.5934 | 0.0815 |
| pstatus | 0.02567 | −0.12743 | 0.09857 | 0.03471 | −0.06984 | −0.02377 | −0.4038 | 0.25312 |
| medu | 0.02069 | 0.43536 | −0.04774 | 0.01269 | 0.05549 | −0.01142 | −0.02168 | 0.02357 |
| fedu | 0.021 | 0.48458 | −0.12044 | −0.05813 | −0.02211 | −0.00722 | −0.08667 | −0.01257 |
| traveltime | 0.06977 | −0.07899 | 0.08823 | −0.42754 | 0.05173 | −0.00324 | 0.09923 | 0.11188 |
| studytime | −0.15099 | −0.12672 | 0.32608 | −0.05097 | 0.07369 | 0.01751 | 0.07721 | 0.21671 |
| failures | 0.03835 | −0.14285 | −0.09268 | 0.02918 | 0.30349 | 0.11095 | −0.12298 | −0.1072 |
| schoolsup | 0.00016 | −0.03632 | −0.01282 | 0.00861 | −0.05149 | −0.02255 | 0.00063 | 0.06034 |
| famsup | −0.0004 | 0.05622 | 0.4013 | −0.08219 | 0.13135 | 0.15243 | −0.18615 | −0.20652 |
| paid | 0.10079 | −0.07894 | 0.53166 | 0.02635 | −0.04372 | −0.03044 | 0.02374 | −0.10245 |
| activities | −0.03136 | 0.1021 | −0.13864 | −0.13376 | 0.04747 | 0.03055 | −0.0209 | 0.65063 |
| nursery | −0.08674 | 0.0782 | 0.11152 | −0.06779 | 0.08835 | 0.17989 | 0.38605 | −0.06069 |
| higher | 0.01206 | 0.08649 | 0.15443 | −0.06712 | −0.32819 | −0.05169 | 0.12033 | 0.10186 |
| internet | 0.05957 | −0.02626 | 0.1502 | 0.3249 | 0.20655 | −0.07449 | −0.08887 | 0.20321 |
| romantic | −0.09428 | 0.0183 | 0.07294 | −0.05578 | 0.56248 | 0.02087 | 0.06626 | 0.09132 |
| famrel | −0.10208 | −0.07048 | 0.06892 | 0.0183 | −0.02425 | 0.53352 | 0.11173 | 0.06412 |
| freetime | 0.12496 | −0.01853 | −0.04392 | 0.11156 | −0.00742 | 0.41924 | 0.00499 | 0.15355 |
| goout | 0.26257 | −0.03328 | 0.041 | 0.12816 | 0.00752 | 0.16957 | 0.05586 | 0.19743 |
| dalc | 0.37671 | 0.02768 | 0.07301 | −0.05661 | −0.04551 | −0.04509 | 0.01312 | −0.05814 |
| walc | 0.40238 | 0.01466 | 0.03822 | −0.04011 | −0.06415 | −0.08516 | −0.00015 | −0.01715 |
| health | −0.04072 | 0.13569 | −0.01753 | −0.18946 | 0.03589 | 0.44102 | −0.10537 | −0.31336 |
| absences | 0.05341 | 0.08776 | −0.01319 | −0.01697 | 0.38273 | −0.17025 | 0.07009 | 0.01427 |

Given the scenarios with consistent feature distribution, we selected eight variables that contributed the most to the principal components as the input features of the model, namely walc, fedu, paid, address, romantic, famrel, famsize, and activities. The source domain still adopts the eight features selected from previous scenarios for scenarios with inconsistent features, while the target domain adopts the new eight features. They are dalc, absences, medu, goout, higher, freetime, studytime and internet. We visualize these two groups of features, use PCA to reduce the dimension to three-dimensional space, and check the distribution of data points. Figure 3 respectively shows the distribution of the features of the source domain and the target domain in the three-dimensional space. To ensure the domain invariant features in the transfer process, we also conducted a pairwise correlation analysis on a total of 16 features and tested the significance of the correlation results. The results are shown in Table 3.

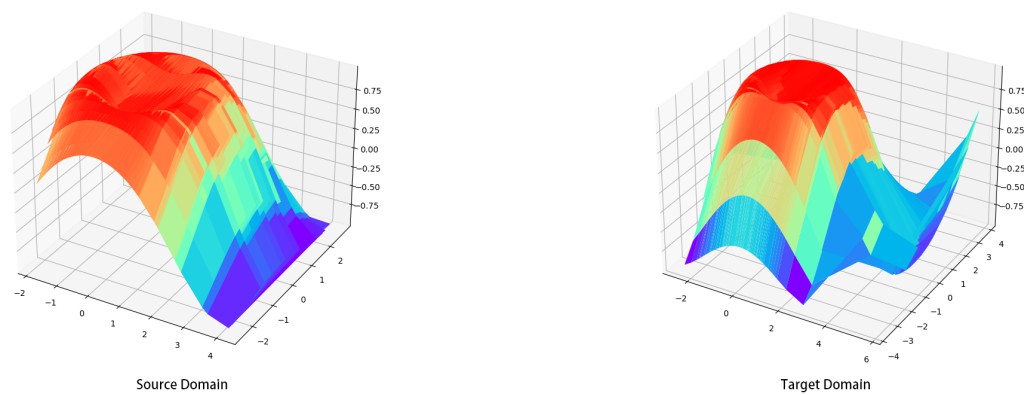

**Figure 3.** Visualization results of features of the source domain and target domain in three-dimensional space.

**Table 3.** Results of significance test of pairwise correlation

| Variables | (1) | (2) | (3) | (4) | (5) | (6) | (7) | (8) | (9) | (10) | (11) | (12) | (13) | (14) | (15) | (16) |
|---|---|---|---|---|---|---|---|---|---|---|---|---|---|---|---|---|
| (1) address | 1 | | | | | | | | | | | | | | | |
| (2) famsize | −0.104 | 1 | | | | | | | | | | | | | | |
| | −0.302 | | | | | | | | | | | | | | | |
| (3) fedu | −0.009 | 0.112 | 1 | | | | | | | | | | | | | |
| | −0.931 | −0.268 | | | | | | | | | | | | | | |
| (4) paid | −0.096 | 0.014 | 0.091 | 1 | | | | | | | | | | | | |
| | −0.343 | −0.888 | −0.366 | | | | | | | | | | | | | |
| (5) activities | −0.125 | 0.026 | 0.104 | 0.101 | 1 | | | | | | | | | | | |
| | −0.214 | −0.798 | −0.305 | −0.32 | | | | | | | | | | | | |
| (6) romantic | −0.085 | 0.098 | 0 | 0.017 | 0.058 | 1 | | | | | | | | | | |
| | −0.401 | −0.331 | −0.996 | −0.864 | −0.57 | | | | | | | | | | | |
| (7) famrel | 0.236 * | −0.155 | −0.101 | 0.153 | 0.024 | −0.103 | 1 | | | | | | | | | |
| | −0.018 | −0.123 | −0.318 | −0.129 | −0.812 | −0.306 | | | | | | | | | | |
| (8) walc | 0.078 | −0.177 | 0.066 | 0.03 | −0.098 | 0.220 * | −0.15 | 1 | | | | | | | | |
| | −0.441 | −0.078 | −0.516 | −0.768 | −0.334 | −0.028 | −0.137 | | | | | | | | | |
| (9) medu | −0.165 | −0.096 | 0.023 | −0.134 | −0.013 | 0.026 | 0.079 | 0.056 | 1 | | | | | | | |
| | −0.1 | −0.34 | −0.82 | −0.184 | −0.899 | −0.8 | −0.437 | −0.578 | | | | | | | | |
| (10) studytime | −0.071 | 0.005 | 0.11 | 0 | 0.106 | −0.024 | 0.054 | 0.039 | 0.092 | 1 | | | | | | |
| | −0.482 | −0.96 | −0.278 | −1 | −0.294 | −0.81 | −0.596 | −0.699 | −0.363 | | | | | | | |
| (11) higher | −0.05 | 0.208 * | 0.015 | −0.19 | 0.014 | 0.067 | −0.159 | −0.164 | 0.253 * | 0.1 | 1 | | | | | |
| | −0.62 | −0.038 | −0.883 | −0.058 | −0.887 | −0.508 | −0.114 | −0.102 | −0.011 | −0.321 | | | | | | |
| (12) internet | −0.05 | 0.094 | 0.151 | −0.058 | −0.017 | 0.189 | −0.009 | −0.206 * | 0.11 | 0.115 | 0.148 | 1 | | | | |
| | −0.623 | −0.353 | −0.134 | −0.569 | −0.864 | −0.06 | −0.931 | −0.04 | −0.278 | −0.256 | −0.141 | | | | | |
| (13) freetime | −0.024 | −0.029 | −0.008 | 0.104 | 0.075 | −0.05 | −0.019 | −0.034 | 0.032 | −0.278 * | −0.198 * | −0.064 | 1 | | | |
| | −0.815 | −0.771 | −0.935 | −0.304 | −0.461 | −0.623 | −0.855 | −0.736 | −0.752 | −0.005 | −0.048 | −0.528 | | | | |
| (14) goout | −0.136 | 0.069 | −0.142 | 0.102 | −0.029 | −0.083 | −0.015 | −0.114 | 0.07 | −0.08 | 0.182 | −0.055 | 0.153 | 1 | | |
| | −0.177 | −0.493 | −0.159 | −0.312 | −0.776 | −0.413 | −0.88 | −0.26 | −0.49 | −0.43 | −0.07 | −0.585 | −0.129 | | | |
| (15) dalc | −0.025 | −0.166 | 0.015 | −0.017 | −0.056 | −0.093 | −0.138 | 0.081 | 0.036 | 0.014 | 0.055 | −0.038 | 0.093 | 0.228 * | 1 | |
| | −0.804 | −0.098 | −0.88 | −0.869 | −0.583 | −0.356 | −0.172 | −0.423 | −0.719 | −0.892 | −0.587 | −0.71 | −0.358 | −0.022 | | |
| (16) absences | 0.019 | 0.045 | 0.049 | 0.142 | 0.069 | −0.164 | 0.036 | −0.189 | −0.004 | 0.065 | −0.043 | −0.157 | −0.225 * | 0.104 | 0.05 | 1 |
| | −0.848 | −0.653 | −0.626 | −0.158 | −0.496 | −0.104 | −0.723 | −0.059 | −0.965 | −0.52 | −0.674 | −0.118 | −0.024 | −0.301 | −0.624 | |

[1]* shows that the correlation between the two variables is significant.

It can be seen from Figure 3 that the feature spaces of the source domain and the target domain are inconsistent, so the two datasets are heterogeneous in features, which guarantees our hypothesis. At the same time, to ensure the transfer of valuable knowledge in the transfer process, the correlation between features is analyzed. From Table 3, there is a specific correlation between the features. The first row of each feature in Table 3 represents the Pearson correlation coefficient between variables, and the second row represents the *p*-value of the significance hypothesis. From the significance test results, there are significantly associated features between the two domains, which ensures the extraction of domain invariant features in the process of domain transfer and the effectiveness of transfer learning.

For the scenario with the same data scale, both the source domain and the target domain used the data of 200 students. In the case of an inconsistent data scale, the source domain uses the data of 200 students, and the target domain uses the data of 80 students.

From the results, we can see that both the source domain and the target domain can make classification predictions when they are not trained. The accuracy is 0.5, which accords with the common sense of binary classification. The accuracy of the domain classifier in each scene is between 0.5 and 0.6, which indicates that the machine can not accurately distinguish the features of the source domain or the target domain, displaying that knowledge has transferred.

## 5. Results and Discussion

In the experiment, we use the logistic regression model to classify the students' academic performance in binary, the network structure is shown in Figure 1, and we use the federated transfer learning framework based on heterogeneous domain adaptation defined by this paper.

In each scenario, we conducted three pieces of training, namely, training only for the source domain, training only for the target domain, and transfer training. We give the classification accuracy of the source domain and target domain under each training mode

and give the classification accuracy of the domain under the transfer training mode. The results are shown in Table 4.

**Table 4.** Training accuracy results of datasets

|  | Source Only Training | | Target Only Training | | Domain Adaptation training | | |
|---|---|---|---|---|---|---|---|
|  | **Source** | **Target** | **Source** | **Target** | **Source** | **Target** | **Domain** |
| Scenario 1 | 0.9045 | 0.5175 | 0.545 | 0.8775 | 0.92 | 0.87 | 0.6 |
| Scenario 2 | 0.9045 | 0.505 | 0.5 | 0.5 | 0.9035 | 0.875 | 0.55 |
| Scenario 3 | 0.75 | 0.5045 | 0.5 | 0.55 | 0.8455 | 0.905 | 0.545 |
| Scenario 4 | 0.75 | 0.5 | 0.45 | 0.475 | 0.8035 | 0.85 | 0.6 |

We can find that, in Scenarios 2–4, the results of target-only training are generally a little worse than source-only training due to the difference in the data size of the source and target domains and the different contributions of the included features to the principal components. Moreover, in Scenario 1, after using domain adaptation training, the accuracy results are not much different from the results of individual training in each domain, which indicates that using domain adaptation training will not improve the model effect in this case. In Scenario 2, after using domain adaptation, the classification accuracy of the target domain is greatly enhanced. In Scenario 3, both the source domain's classification accuracy and the target domain have dramatically increased. However, in Scenario 4, the classification accuracy of the source domain and the target domain increased less than that of Scenario 3.

The result analysis shows that the domain adaptation model has little effect when the feature distribution and scale of the source domain and the target domain are consistent. However, when one party's data scale is small, it can benefit from federated learning. Especially when the feature distribution of both parties is inconsistent, both parties can learn new knowledge from each other and improve the classification accuracy. In Scenario 4, although we project the source and target domains into a common feature subspace, since the source and target domains have different feature distributions and data scales, the training knowledge on the source domain is directly applied to the target domain, and the resulting effect will be inferior to Scenario 2 and Scenario 3.

From the experimental results, the federated transfer learning framework designed by us can effectively solve the problem of federated learning with small unilateral data and missing unilateral features in students' grades classification, and all participants can obtain new knowledge from each other while maintaining privacy. Experiments show that applying a federated learning framework can solve the problem of data islands in educational data mining, which can be used for reference in the field of intelligent education.

The study has potential limitations. Since the experimental part of this paper only verifies the data exchange between two different education platforms, and the data differences generated by different education platforms may be large, how to transfer privacy protection knowledge between more online education platforms still needs further exploration.

## 6. Conclusions

This paper proposes a framework based on federated transfer learning for students' grades classification with privacy protection. By introducing feature extractors and domain classifiers, the framework matches the feature distribution of all parties in collaborative learning in some feature spaces. It uses homomorphic encryption and random masking to ensure data privacy during transmission. The experimental results show that both parties with different data feature spaces and inconsistent data scales can effectively transfer knowledge under this framework. It can help the target data parties with missing data labels, or fewer feature data train the students' grades classification model and improve the generalization ability and classification effect of the students' grades classification model of the primary data participants.

In future work, we will further study privacy protection in the framework. This includes studying privacy protection in the federated learning framework in the presence of malicious participants, and more efficient feature extraction methods. We are committed to expanding federated learning technology to more fields and exploring more effective federated learning methods. These methods can include studying and further improving this method's communication efficiency and effectiveness in more complex and asynchronous federated learning systems.

**Author Contributions:** Funding acquisition, B.X.; methodology, S.Y. and B.X.; conceptualization, S.Y.; software, S.Y.; validation, S.Y. and B.X.; writing, review and editing, S.Y., S.L. and Y.D.; supervision, S.L. All authors have read and agreed to the published version of the manuscript.

**Funding:** This research was funded by the National Natural Science Foundation of China, grant (U1811261), and the Fundamental Research Funds for the Central Universities of China, grant (N2116019), and the Liaoning Natural Science Foundation, grant (2022-MS-119).

**Institutional Review Board Statement:** Not applicable.

**Informed Consent Statement:** Not applicable.

**Data Availability Statement:** Not applicable.

**Conflicts of Interest:** The authors declare no conflict of interest.

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
