# Peer review of "A Federated Transfer Learning Framework Based on Heterogeneous Domain Adaptation for Students’ Grades Classification"

_applsci, doi:10.3390/app122110711_

Round 1
Reviewer 1 Report
The paper analyses the topic of student performance assessment with an original and in many ways innovative approach. The theoretical part of the paper is clear and comprehensive, and the models are convincingly described. One cannot, however, fail to notice a weakness in the presentation of the empirical results, which appears too reductive and does not allow one to fully appreciate the quality of the research conducted. This is a part that should be strengthened in terms of a better and broader presentation. The model based on logistic regression should be better described and the relevant tables should be included in the text. With respect to the tabulation of the results, the part concerning the goodness of fit should be made more explicit. The overly synthetic empirical part risks compromising the overall quality of the work, which for the first four paragraphs was of an appreciably high standard. The research as a whole is very good, but this too reduced empirical part risks detracting from its value.
Author Response
Dear Editors and reviewers:
Re: Manuscript ID: applsci-1936154 and Title: A Federated Transfer Learning Framework Based on Heterogeneous Domain Adaptation for Students’ Grades Classification
Thank you for your precious comments and advice. Those comments are all valuable and very helpful for revising and improving our paper, as well as the important guiding significance to our researches. We have studied comments carefully and have made correction which we hope meet with approval. The main corrections in the paper and the responds to the reviewer’s comments are as flowing:
Reviewer #1:
Point 1:The paper analyses the topic of student performance assessment with an original and in many ways innovative approach. The theoretical part of the paper is clear and comprehensive, and the models are convincingly described.
Response 1: Thank you for your summary. We really appreciate your efforts in reviewing our manuscript. We have revised the manuscript accordingly. Our point-by-point responses are detailed below.
Point 2: One cannot, however, fail to notice a weakness in the presentation of the empirical results, which appears too reductive and does not allow one to fully appreciate the quality of the research conducted. This is a part that should be strengthened in terms of a better and broader presentation. The model based on logistic regression should be better described and the relevant tables should be included in the text.
Response 2:We are grateful for the suggestion. In the experiences section, we have supplemented more detailed parameter descriptions of the model, such as the learning rate, the number of layers of the DNN, etc. and put the table of experimental results in the text.
Point 3: With respect to the tabulation of the results, the part concerning the goodness of fit should be made more explicit.
Response 3:We are very grateful to your comments for the manuscript. To be more clearly and in accordance with the reviewer concerns, we have added a more detailed interpretation regarding experimental results on page 12-13.
Thank you for your careful review. We really appreciate your efforts in reviewing our manuscript during this unprecedented and challenging time. We wish good health to you, your family, and community. Your careful review has helped to make our study clearer and more comprehensive.
Sincerely.

Reviewer 2 Report
The article is devoted to describing a solution to the common problem of aggregation of educational data from different sources, which is important for educational data mining and machine learning. The article describes a method of secure federated learning, aimed at allowing different parties to exchange private data in order to improve their machine-learning models without disclosing sensitive data to each other.
The article's chief strength is in its method section. However, the article has several problems, which should be fixed to increase the article's impact.
1. While the article has a decent introduction and review of related works, the same cannot be said about the discussion of the results (section 5) which is brief and sketchy. There is no comparison of the authors' results with published results in the previous literature, which can lead to conclusions about which information in the article is new and which confirms previous knowledge. Also, there's no analysis of statistical significance of the results. The results could also be discussed in more depth (e.g., why target-only training gave consistently bad results for the target dataset in Scenarios 2-4).
2. In the introduction the authors claim trying to solve the problem of mining educational data that "are stored in different schools and on various online education platforms", however the experimental data was taken from two Portuguese schools, and to simulate different features the authors artificially limited the number of selected features in some scenarios. It is not conclusive how this can be generalized to exchanging data between different online educational platforms (there's no analysis of the actual difference between online platforms and whether the experimental conditions are close to it) and different student populations (e.g. different countries, different regions of federative countries, etc.). We don't even know how close to each other these two schools are located and how similar their administration is, which is important for interpreting the results of experiments. I strongly suggest adding a section about limitations of this study and propose a plan of further study to determine the boundaries of its applicability (when it makes sense to use federated learning between two educational institutions, and when the difference in students and/or administration makes it not viable?).
3. The article claims to describe a framework, but it only really describes a theoretical framework - not its implementation, which was used in the experiment. We get no information about the actual programming code framework developed to test the authors' hypotheses and if this code is available under open source license or at request. This is important both to ensure reproducibility of the authors' results and to give other researchers the ability to use and study this framework further. If the authors choose to not share their program code, the rationale for doing it can be provided.
Also, there are technical errors, like
1. At pages 9 and 10, Theorem 2 and Theorem 3 look identical, as well as their proofs.
2. In the line 179 the authors state "The network we use is a deep neural network." This is a very broad definition. Consider defining more precisely the class of artificial neural networks your framework is supposed to work with. Figure 3 shows the structure of the neural network used in the experiment, but it is shown far too late in the article; consider moving it to the method section and describing it more. The applicability of the proposed framework to other kinds of neural networks can be discussed in the "limitations" section.
3. In lines 37-38 the authors state "There is an unbreakable barrier between these data sources" but the article is devoted to breaking this barrier. So it must be breakable. If it really were unbreakable, there would be no use in trying to break it.
The article also requires serious English editing. It contains a lot of poorly formulated phrases and sentences like "many industries have data-limited and poor quality, which is not enough to...", "In this section, we mainly introduce applying this method to computer-aided education", "Under this framework, all participants can benefit. Moreover, after analysis, this framework is safe." etc.
Author Response
Dear Editors and reviewers:
Re: Manuscript ID: applsci-1936154 and Title: A Federated Transfer Learning Framework Based on Heterogeneous Domain Adaptation for Students’ Grades Classification
Thank you for your precious comments and advice. Those comments are all valuable and very helpful for revising and improving our paper, as well as the important guiding significance to our researches. We have studied comments carefully and have made correction which we hope meet with approval. The main corrections in the paper and the responds to the reviewer’s comments are as flowing:
Reviewer #2:
Point 1: There is no comparison of the authors' results with published results in the previous literature.
Response 1:We are grateful for the suggestion. As suggested by the reviewer, we tried to compare the model proposed in this paper with the previous models, but we found that the datasets selected by the methods in the previous literature are not the same, so they do not have the conditions and value of direct comparison. And the focus of the experiments in this paper is to verify the effectiveness of transfer learning. We give two control groups for each scenario when designing the experiment, namely source-only training and target-only training. The experimental results also show that the transfer learning mechanism is effective in our framework.
Point 2: there's no analysis of statistical significance of the results. The results could also be discussed in more depth (e.g., why target-only training gave consistently bad results for the target dataset in Scenarios 2-4).
Response 2:We deeply appreciate the reviewer’s suggestion. According to the reviewer’s comment, we have added a more detailed interpretation regarding experimental results on page 12-13.
Point 3: I strongly suggest adding a section about limitations of this study and propose a plan of further study to determine the boundaries of its applicability (when it makes sense to use federated learning between two educational institutions, and when the difference in students and/or administration makes it not viable?).
Response 3:Thank you for your precious comments and advice. Admittedly, since the experimental part of this manuscript only verified the data exchange between two different education platforms, and the data generated by different education platforms may be quite different, but the time and research funding we can now invest are limited, so how to transfer knowledge about privacy protection between more online education platforms still needs further exploration. And we have added relevant explanations in the 'Results and Discussion' section.
Point 4: The article claims to describe a framework, but it only really describes a theoretical framework - not its implementation, which was used in the experiment. We get no information about the actual programming code framework developed to test the authors' hypotheses and if this code is available under open source license or at request. This is important both to ensure reproducibility of the authors' results and to give other researchers the ability to use and study this framework further. If the authors choose to not share their program code, the rationale for doing it can be provided.
Response 4:Thank you for your comments. All the experimental data of this manuscript are from the source code we run. However, due to the nonstandard storage of our source code, we cannot open source it yet, we will sort out the source code files as soon as possible, and our source code is available upon request.
Point 5: At pages 9 and 10, Theorem 2 and Theorem 3 look identical, as well as their proofs.
Response 5:Thank you for your careful review. We apologize for the mistakes in the manuscript and also carefully checked the entire manuscript for typographic, grammatical and formatting errors.
Point 6: In the line 179 the authors state "The network we use is a deep neural network." This is a very broad definition. Consider defining more precisely the class of artificial neural networks your framework is supposed to work with.
Response 6:Thank you for your careful review. The neural network we use is actually a simple deep neural network (DNN) with three layers, not a convolutional neural network or a recurrent neural network. In fact, we mainly optimize knowledge transfer in the manuscript, but not the neural network used for feature extraction.
Point 7: Figure 3 shows the structure of the neural network used in the experiment, but it is shown far too late in the article; consider moving it to the method section and describing it more.
Response 7:Thank you for your precious comments and advice. We've moved the Figure 3 to the method section and added a more detailed interpretation regarding Figure 3 on page 5.
Point 8: The applicability of the proposed framework to other kinds of neural networks can be discussed in the "limitations" section.
Response 8:Thank you for your precious comments and advice. In our proposed framework, we use a simple deep neural network to extract features. We do not use other kinds of neural networks, but we believe that there will be new improvements in feature extraction. Since we mainly improved the knowledge transfer problem in the manuscript, we did not do much discussion on the feature extraction module, but we can further explore the applicability of other kinds of neural networks in the framework proposed in this manuscript in future work.
Point 9: In lines 37-38 the authors state "There is an unbreakable barrier between these data sources" but the article is devoted to breaking this barrier. So it must be breakable. If it really were unbreakable, there would be no use in trying to break it.
Response 9:Thank you for your careful review. I'm sorry that we made mistakes in our expression, and we have revised this sentence.
Point 10: The article also requires serious English editing. It contains a lot of poorly formulated phrases and sentences like "many industries have data-limited and poor quality, which is not enough to...", "In this section, we mainly introduce applying this method to computer-aided education", "Under this framework, all participants can benefit. Moreover, after analysis, this framework is safe." etc.
Response 10:We agree with the comment, We are very sorry for the mistakes in this manuscript and inconvenience they caused in your reading. We rechecked the grammar and re-wrote the sentences with errors in the revised manuscript.
Thank you for your careful review. We really appreciate your efforts in reviewing our manuscript during this unprecedented and challenging time. We wish good health to you, your family, and community. Your careful review has helped to make our study clearer and more comprehensive.
Sincerely.

Reviewer 3 Report
1. In the introduction, it is recommended to present how the paper is structured.
2. The results of some tables (eg. 3 or/and 4) can also be represented in a figure.
3. Some more up-to-date references (last 2-3 years) can be added to the manuscript.
4. Related work should be improved.
5. The overall organization and structure of the paper should be improved.
Author Response
Dear Editors and reviewers:
Re: Manuscript ID: applsci-1936154 and Title: A Federated Transfer Learning Framework Based on Heterogeneous Domain Adaptation for Students’ Grades Classification
Thank you for your precious comments and advice. Those comments are all valuable and very helpful for revising and improving our paper, as well as the important guiding significance to our researches. We have studied comments carefully and have made correction which we hope meet with approval. The main corrections in the paper and the responds to the reviewer’s comments are as flowing:
Reviewer #3:
Point 1: In the introduction, it is recommended to present how the paper is structured.
Response 1:We are grateful for the suggestion. As suggested by the reviewer, we have added more details of how the paper is structured in the introduction.
Point 2: The results of some tables (eg. 3 or/and 4) can also be represented in a figure.
Response 2:Thank you for your comments, but we think it is better not to represent Table 3 and 4 in figures, because the floating point number in Table 3 and 4 can help readers understand the results of significance test of pairwise correlation more accurately.
Point 3: Some more up-to-date references (last 2-3 years) can be added to the manuscript.
Response 3:We are grateful for the suggestion. We have read more literature and cited appropriate literature into the manuscript.
Point 4: Related work should be improved.
Response 4:We are very grateful to your comments for the manuscript. We adjusted the Related work and cited more literatures in the last two years.
Point 5:The overall organization and structure of the paper should be improved.
Response 5:We agree with your comment, and we have adjusted the position of some paragraphs and tables to make the manuscript easier to read.
Thank you for your careful review. We really appreciate your efforts in reviewing our manuscript during this unprecedented and challenging time. We wish good health to you, your family, and community. Your careful review has helped to make our study clearer and more comprehensive.
Sincerely.
